# Asymmetric Bethe Ansatz

Steven G. Jackson[1], Gregory E. Astrakharchik[2], and Maxim Olshanii[3*]

**1** Department of Mathematics, University of Massachusetts Boston, Boston Massachusetts 02125, USA
**2** Departament de Física, Universitat Politècnica de Catalunya, E08034 Barcelona, Spain
**3** Department of Physics, University of Massachusetts Boston, Boston Massachusetts 02125, USA
maxim.olchanyi@umb.edu

November 25, 2023

## Abstract

The recently proposed exact quantum solution for two $\delta$-function-interacting particles with a mass-ratio $3\!:\!1$ in a hard-wall box [**Y. Liu, F. Qi, Y. Zhang and S. Chen, iScience 22, 181 (2019)**] seemingly violates the conventional necessary condition for a Bethe Ansatz integrability for a system of semitransparent $\delta$-function mirrors: *if two mirrors of a Bethe-Ansatz-solvable model cross at a dihedral angle $\pi/$(odd number), these mirrors must be assigned equal coupling constants.* In our article, we find a way to relax this condition: it turns out that one can take a conventional integrable system and replace some of its semi-transparent mirrors by perfectly reflecting ones. The latter set must be represented by the mirrors of a reflection subgroup of the symmetry group of the conventional system. This subgroup is *not* required to be symmetric with respect to the symmetries original system, hence the proposed name for the method: *Asymmetric Bethe Ansatz* (**ABA**). We show that the exact solution of the Liu-Qi-Zhang-Chen problem is a particular instance of the ABA.

# 1 Introduction

## 1.1 Multidimensional kaleidoscopes

The concept of a *generalized kaleidoscope* is the key to understanding the mathematical nature of Bethe Ansatz solvability for some systems of $\delta$-function-interacting particles [1]. It is a generalization of a conventional notion of a kaleidoscope to kaleidoscopic systems of, generally, semitransparent mirrors.

A conventional kaleidoscope is a system non-transparent mirrors that is invariant under all reflections generated by its own mirrors. As a consequence of this invariance, images of objects in a kaleidoscope are not distorted at the junctions between the mirrors, rendering these junctions invisible for an observer.

Transformations of space generated by all possible sequences of reflections about mirrors of a kaleidoscope form a *reflection group* [2].

All indecomposable (to products of reflection subgroups) reflection groups are known. They are catalogued using the so-called Coxeter diagrams [3]; the latter are shown in Figs. 1-2. Decomposable reflection groups can be factorized onto products of the indecomposable ones.

Figure 1 lists the so-called indecomposable *finite* reflection groups. For a finite reflection group, all of its mirrors cross the origin. The mirrors divide the space onto the infinite volume *chambers*. Each of the chambers is connected to any other via an element of the corresponding reflection group. Reflections about chamber's mirrors can be used to generate the whole reflection group.

For an indecomposable finite reflection group, the number of mirrors that form its chamber equals the number of spatial dimensions of the space in which the group operates: as a result, the dihedral angles between its mirrors fully determine the corresponding kaleidoscope. In Fig. 1, circles represent the mirrors of a chamber. The angle between two mirrors whose circles are not directly connected by an edge is $\pi/2$. Edges with no label correspond to an angle $\pi/3$ between the mirrors. Otherwise, an index $m$ on an edge would give an angle $\pi/m$.

Figure 2 catalogues the so-called indecomposable *affine* reflection groups. There, the mirrors divide the space into finite-volume *alcoves*. The number of mirrors that bound an alcove exceeds the number of the spatial dimensions by one. As the result alcoves are closed simplices. Similarly to the case of indecomposable finite reflection groups, each of the alcoves is again connected to any other via an element of the corresponding reflection group; reflections about the walls of any given alcove generate the full reflection group.

In the affine case, mirrors are arranged in space into periodic series. Each affine group has a finite partner (the converse is not generally true): a union of images of an alcove produced by the corresponding finite group forms an elementary periodicity cell, whose periodic translations tile the space.

Figure 2 lists the indecomposable affine reflection groups. Each Coxeter diagram encodes the dihedral angles between the mirrors of the alcove of the group. Here, notations are identical to the ones for the finite groups. However, unlike in the finite case, dihedral angles between the mirrors of an alcove determine the corresponding kaleidoscope only up to an arbitrary dilation factor.

Decomposable reflection groups produce disconnected Coxeter diagrams, one connected sub-graph for each indecomposable factor (with factors being represented by either an indecomposable finite or an indecomposable affine reflection group). For the decomposable groups, the notion of a chamber (in case of an infinite volume) or alcove (for finite volume kaleidoscopes) can be preserved without modification. Note however that when one or

more of the indecomposable factors of the decomposable group are affine, the Coxeter diagram and the dihedral angles between the chamber/alcove mirrors it codifies fix the corresponding kaleidoscope only up to a set of arbitrary dilation factors, one for each indecomposable affine factor of the decomposable reflection group in question.

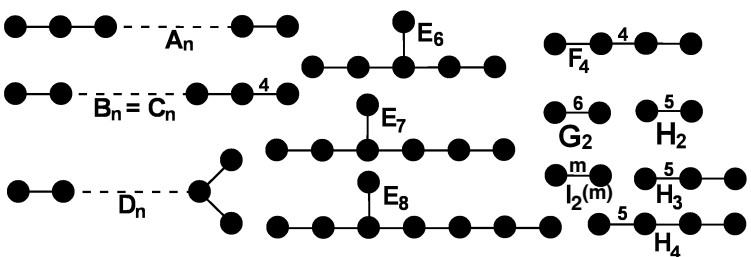

Figure 1: **A full list of indecomposable finite reflection groups, encoded by the Coxeter diagrams**. For $A_n$, $n = 0, 1, 2, \ldots$. $A_0$ coincides with the trivial group (i.e the group whose only element is the identity). For $B_n = C_n$, $n = 2, 3, 4, \ldots$. For $D_n$, $n = 4, 5, 6, \ldots$. For $I_2(m)$, $m = 5, 7, 8, 9, \ldots$.

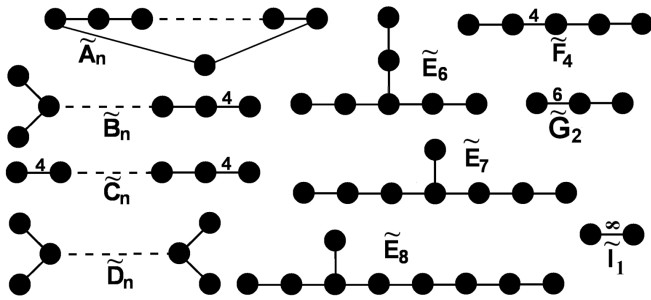

Figure 2: **A full list of indecomposable affine reflection groups, encoded by the Coxeter diagrams**. For $\tilde{A}_n$, $n = 2, 3, 4, \ldots$. Here $\tilde{I}_1$ corresponds to two parallel mirrors as an alcove. Note that in some texts, $\tilde{A}_1$ is used instead of $\tilde{I}_1$. For $\tilde{B}_n$, $n = 3, 4, 5, \ldots$. For $\tilde{C}_n$, $n = 2, 3, 4, \ldots$. For $\tilde{D}_n$, $n = 4, 5, 6, \ldots$. For $I_2(m)$, $m = 5, 7, 8, 9, \ldots$.

## 1.2 Gaudin's generalized kaleidoscopes

In his classic book [4], Michel Gaudin proposed (in Sec. 5.2) a model that he called a *generalized kaleidoscope*. Consider a standard kaleidoscope. Let us replace some of its mirrors by $\delta$-function potentials $g_{\mathbf{n}, \mathbf{r}_0} \delta(\mathbf{n} \cdot (\mathbf{r}) - \mathbf{r}_0)$, with the coupling constants $g_{\mathbf{n}, \mathbf{r}_0}$ chosen in such a way that the resulting system of semitransparent mirrors still respects the reflection group of the original kaleidoscope. Here, $\mathbf{n}$ is a unit normal to the mirror, $\mathbf{r}_0$ is a point on the mirror, and $g_{\mathbf{n}, \mathbf{r}_0}$ is the strength of the $\delta$-function potential. The remaining non-transparent mirrors can be naturally reinterpreted as $\delta$-function potentials of an infinite strength. A generalized kaleidoscope problem is a problem of finding the eigenstates and eigenenergies for a multidimensional quantum particle moving in such potential. According to Ref. [1], any generalized kaleidoscope is solvable using Bethe Ansatz.

The generalized kaleidoscope model is a natural generalization of the three textbook Bethe-Ansatz-solvable, empirically relevant models: $N$ $\delta$-potential-interacting spinless bosons [5] or spin-1/2 fermions [6,7] on a ring (reflection group $\tilde{A}_{N-1}$), and $N$ $\delta$-potential-interacting bosons in a hard-wall box (reflection group $\tilde{C}_N$) [7].

One particular consequence of the reflection symmetry of any generalized kaleidoscope is as follows:

**Integrability Condition 1 (Standard necessary condition for Bethe Ansatz integrability)** *If a system of $\delta$-function mirrors is Bethe-Ansatz-integrable, then any two of its $\delta$-function that cross at a dihedral angle $\pi/$(odd number) carry the same coupling constant.*

Indeed, for two mirror reflections $\hat{\mathcal{T}}_1$ and $\hat{\mathcal{T}}_2$ whose mirrors cross at a dihedral angle $\pi/m$, their composition

$$\mathcal{T}_1\mathcal{T}_2 = \mathcal{R}_{\frac{2\pi}{m}}$$

is a rotation by $\frac{2\pi}{m}$ in the plane spanned by the normals $\mathbf{n}_1$ and $\mathbf{n}_2$ to the $\hat{\mathcal{T}}_1$ and $\hat{\mathcal{T}}_2$ mirrors respectively, in the direction from $\mathbf{n}_1$ to $\mathbf{n}_2$. Under these conventions, $\mathbf{n}_1$ and $\mathbf{n}_2$ are related as

$$\mathbf{n}_1 = \mathcal{R}_{-\frac{\pi}{m}}\mathbf{n}_2 .$$

Applying the $\mathcal{T}_1\mathcal{T}_2$ transformation $\frac{m-1}{2}$ times, one gets

$$(\mathcal{T}_1\mathcal{T}_2)^{\frac{m-1}{2}} = \mathcal{R}_{\pi-\frac{\pi}{m}} .$$

Applying this transformation to the normal to the second mirror we get

$$(\mathcal{T}_1\mathcal{T}_2)^{\frac{m-1}{2}}\mathbf{n}_2 = \mathcal{R}_{\pi-\frac{\pi}{m}}\mathbf{n}_2 = -\mathbf{n}_1 .$$

Since both $\hat{\mathcal{T}}_1$ and $\hat{\mathcal{T}}_2$ are mirrors of the kaleidoscope of interest, the corresponding generalized kaleidoscope must be invariant under any element of the reflection group generated these mirrors; in particular, it must be invariant under the transformation $(\mathcal{T}_1\mathcal{T}_2)^{\frac{m-1}{2}}$. This invariance requires that the coupling strengths are equal for the two mirrors under consideration. Notice that the above construction requires $\frac{m-1}{2}$ be integer, and hence $m$ itself be odd. Fig. 3 illustrates our construction for the case $m=5$, in two dimensions.

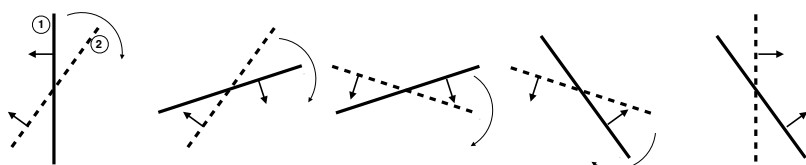

Figure 3: **A two-dimensional illustration to the Integrability Condition 1.**

Recall that above, we assumed that in order to be Bethe-Ansatz-solvable solvable, a generalized kaleidoscope must be symmetric with respect to the reflection group it is generated by. The Asymmetric Bethe Ansatz solvability considered below relaxes this requirement.

## 1.3  Open question: the source of integrability of the Liu-Qi-Zhang-Chen system

In Ref. [8], authors present an exact solution for a quantum system consisting of two $\delta$-interacting particles with a mass ratio 3:1 in a one-dimensional hard-wall box. After

a suitable change of variables, the problem reduces to a motion of a two-dimensional scalar-mass particle in a rectangular well with a width-to-height ratio $1 : \frac{1}{\sqrt{3}}$, divided by a *finite strength* $\delta$-function barrier along its grand diagonal. If the standard Bethe Ansatz had been the source for the solution, both the walls and the barrier would be mirrors of a kaleidoscope. However, in that case, the distribution of the coupling constants—for example a finite coupling on the diagonal and an infinite coupling on the left vertical wall, at $\pi/3$ to each other—would contradict the Integrability Condition 1.

The absence of an explanation for the existence of the exact solution to the Liu-Qi-Zhang-Chen system was our primary motivation for this project.

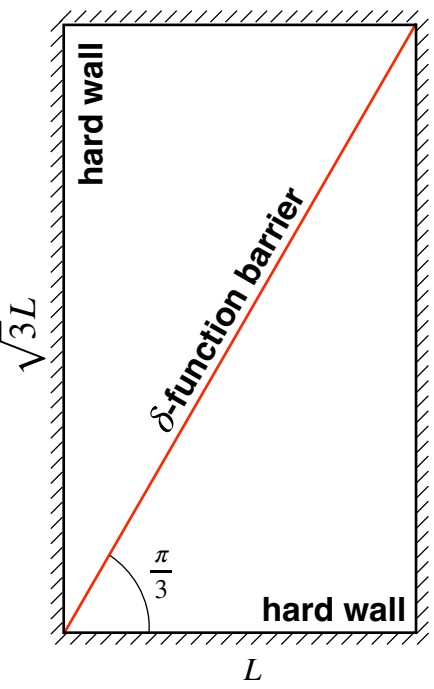

Figure 4: **A rendering of the Liu-Qi-Zhang-Chen system suitable for analysis of its integrability.** A two-dimensional scalar mass quantum particle is assumed. The original system consisted of two one-dimensional $\delta$-interacting particles ($V(x_1 - x_2) = g\delta(x_1 - x_2)$) with masses $m_1$ and $m_2 = 3m_1$, in a hard-wall box of length $L$ [8].

In this paper, our objective is to find the underlying mathematical mechanism that allows for the Liu-Qi-Zhang-Chen solution [8]. We aim to identify a general mathematical phenomenon behind it, and in doing so, to enlarge the standard Bethe Ansatz paradigm.

# 2 Asymmetric Bethe Ansatz

## 2.1 Explaining the Liu-Qi-Zhang-Chen solution and the Asymmetric Bethe Ansatz

We suggest the following explanation for the existence of the exact solution of the Liu-Qi-Zhang-Chen problem [8].

1. Consider a generalized kaleidoscope based on a reflection group $\mathcal{G} = \tilde{G}_2$ (see Fig. 5(a), red and blue lines). Below, we will refer to it as a $\mathcal{G}$-generated generalized kaleidoscope.

Here, the red and blue lines represent mirrors of that kaleidoscope. Two mirrors, each shown with a different color, are allowed to have two different coupling constants without violating the Integrability Condition 1.

2. Now, consider a *reflection subgroup* of $\tilde{G}_2$ represented by $\mathcal{H} = \tilde{A}_1 \times \tilde{A}_1$: the mirrors of the latter are shown as the black dashed lines. Note that generally, such a subgroup is *not symmetric* with respect to the principal group $\mathcal{G}$.

3. Consider the eigenstates of the $\mathcal{G}$-generated generalized kaleidoscope that are odd with respect to each of the mirror reflections that generate $\mathcal{H}$. Such eigenstates will be, at the same time, the eigenstates of another quantum system. To build this system one starts from the $\mathcal{G}$-generated generalized kaleidoscope and replaces the mirrors coinciding with the mirrors of $\mathcal{H}$ with the infinite strength walls. Let us call it a $\mathcal{G}$-$\mathcal{H}$-generated asymmetric generalized kaleidoscope.

   Note that the new coupling constant assignment *does violate* the Integrability Condition 1. Indeed the infinite coupling "black dashed" mirrors and finite coupling "red mirrors" cross at a $\pi/3$ angle.

4. The $\mathcal{H}$-odd eigenstates of the $\mathcal{G}$-generated generalized kaleidoscope form a complete basis in the space of functions that are $\mathcal{H}$-odd. Hence the $\mathcal{H}$-odd eigenstates solve the problem of quantization of the $\mathcal{G}$-$\mathcal{H}$-generated asymmetric generalized kaleidoscope.

The Liu-Qi-Zhang-Chen system can be reinterpreted as an alcove of a $\mathcal{G}$-$\mathcal{H}$-generated asymmetric generalized kaleidoscope, with $\mathcal{G} = \tilde{G}_2$ and $\mathcal{H} = \tilde{A}_1 \times \tilde{A}_1$, for the following assignments

  - The "red" mirrors of the $\mathcal{G}$-$\mathcal{H}$-generated asymmetric generalized kaleidoscope (see Fig. 5) are assigned the coupling constant along the "red" $\delta$-plate of the Liu-Qi-Zhang-Chen problem (Fig. 4).

  - The "blue" mirrors of the $\mathcal{G}$-$\mathcal{H}$-generated asymmetric generalized kaleidoscope (see Fig. 5) are assumed to be transparent (i.e. assigned a zero coupling constant)[1].

  - One of the "alcoves" of the reflection group $\mathcal{H}$ (a part of space bounded by its mirrors, grey rectangle at Fig. 5)(a) is interpreted as the rectangular box Liu-Qi-Zhang-Chen 2D particle is confined to.

This concludes our interpretation of the exact eigenstates of the Liu-Qi-Zhang-Chen system (Fig. 4) as an instance of applicability of an extended, "asymmetric" version of the conventional Bethe Ansatz [1]. We suggest *Asymmetric Bethe Ansatz* (ABA) as the name for this method. In the same vein, we will call the corresponding system of mirrors a *asymmetric generalized kaleidoscope*. Figure 5(b) shows the ground state of the Liu-Qi-Zhang-Chen system.

ABA features a relaxed version of the Integrability Condition 1:

**Integrability Condition 2 (Necessary condition for the Asymmetric Bethe Ansatz integrability)** *If a system of $\delta$-function mirrors is Asymmetric-Bethe-Ansatz-integrable, then any two of its $\delta$-function that cross at a dihedral angle $\pi/$(odd number) carry the same coupling constant, unless one or both mirrors are mirrors of an asymmetric reflection subgroup of the underlying reflection group. Such mirrors can be then assigned an infinite coupling constant.*

---

[1]Interestingly, for this particular choice of the coupling constants, the $\mathcal{G}$-generated generalized kaleidoscope is identical to the one generated by the $\tilde{A}_2$ reflection group ("red" mirrors alone). We would like to stress that the fact that $\mathcal{H} = \tilde{A}_1 \times \tilde{A}_1$ is not a reflection subgroup of $\tilde{A}_2$ has no significance. Here and in general, it is imperative that the group $\mathcal{H}$ is a *reflection subgroup* of the group $\mathcal{G}$ whose eigenstates are used to generate the eigenstates of the $\mathcal{G}$-$\mathcal{H}$-generated asymmetric generalized kaleidoscope.

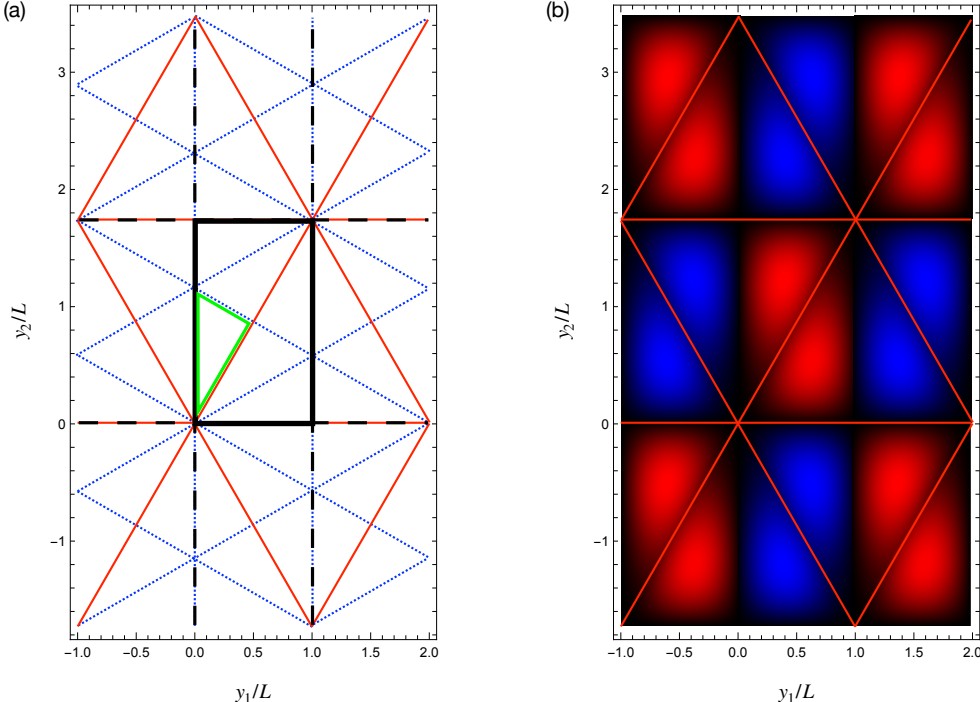

Figure 5: **The Liu-Qi-Zhang-Chen system (Fig. 4) as an alcove of a $\mathcal{G}$-$\mathcal{H}$-generated asymmetric generalized kaleidoscope.** Here, $\mathcal{G} = \tilde{G}_2$, $\mathcal{H} = \tilde{A}_1 \times \tilde{A}_1$. The boundaries of the alcoves of the $\mathcal{G}$ and $\mathcal{H}$ groups are represented by solid green and solid black lines respectively. (a) The $\delta$-function mirrors of the $\mathcal{G}$ group. For the Liu-Qi-Zhang-Chen system, the coupling constant on the "red" mirrors is the same as in the original problem (Fig. 4), and the "blue" coupling constant is set to zero. Black dashed lines are the mirrors of the $\mathcal{H}$ group; the solid-black-walled rectangle is its alcove. (b) Heat-map plot of the ground state wave function of the Liu-Qi-Zhang-Chen problem, computed numerically, positive (negative) values are shown in red (blue) color. According to the Asymmetric Bethe Ansatz protocol, the set of the eigenstates of the problem can be found by identifying a subset of the eigenstates of the $\mathcal{G}$-kaleidiscope (red lines) that are odd with respect to the reflections about the mirrors of $\mathcal{H}$.

## 2.2 Physical manifestations of integrability of the Liu-Qi-Zhang-Chen system

In this Subsection, we will present an example of an empirical manifestation of integrability of the Liu-Qi-Zhang-Chen system. It is common to use the level spacing statistics [9] as an integrability test. However, for the reasons outlined below, in this system, some traces of integrability survive even for large perturbations away from it, invalidating the level statistics method. Instead, we resort to a test for the primary features that deliver validity to the level spacing statistics analysis: the true and avoided crossing between the energy levels plotted as a function of an integrability-unrelated parameter [9].

We choose $\beta \equiv m_2/m_1$ as the integrability controlling parameter, and the coupling constant $g$ (the coefficient in front of the $\delta$-interaction $V(x_1 - x_2) = g\delta(x_1 - x_2)$) as the integrability-unrelated variable. Our primary object of interest is crossings between the levels of the same generic (i.e. present at all $\beta$) symmetry, common to both systems (and for that matter, for all *beta*).

In Fig. 6(a), we show the energy spectrum of the Asymmetric Bethe Ansatz integrable

Liu-Qi-Zhang-Chen system, i.e. for mass ratio of tho particles equal to $\beta = 3$. It is instructive to compare it to the spectrum of non-integrable counterpart shown in Fig. 6(b) where we use mass ratio equal to $\beta = \frac{5}{2} \times (1 + 1/(2 + 1/(3 + 1/(4 + \cdots))))) = 3.5828185668057793\ldots$. This specific choice of $\beta$ is being designed to represent a generic irrational number with manifestly unbounded continued fraction coefficients.

For any value of the mass ratio $\beta$, the system is invariant under a $180°$ rotation about the center of the corresponding rectangular billiard similar to the one represented in Fig. 4. In Fig. 6, red(black) energy lines correspond to the states that are even(odd) under this rotation.

In the absence of interactions ($g = 0$), the spectrum can be found explicitly for any mass ratio $\beta$ and is expressed in terms of two quantum numbers $n_1$ and $n_2$, according to

$$E_{n_1, n_2}^{g=0} = \frac{\beta n_1^2 + n_2^2}{\beta + 1} \mathcal{E}_0 .$$
$$n_1 \geq 1, \, n_2 \geq 1 \tag{1}$$

Here and below, $\mathcal{E}_0 = \frac{\pi^2 \hbar^2}{2\mu L^2}$ denotes the typical energy associated with zero-point fluctuations and $\mu \equiv \frac{m_1 m_2}{m_1 + m_2}$ the reduced mass. The noninteracting ground state energy is $E_{1,1}^{g=0} = \mathcal{E}_0$ independently of the specific value of $\beta$.

For impenetrable particles, $g = \infty$, the energy spectrum of the Liu-Qi-Zhang-Chen system is given by

$$E_{\tilde{n}_1, \tilde{n}_2}^{g=\infty} = \frac{\tilde{n}_1^2 + \tilde{n}_1 \tilde{n}_2 + \tilde{n}_2^2}{3} \mathcal{E}_0 .$$
$$\tilde{n}_2 > \tilde{n}_1 \geq 1 \tag{2}$$

The ground state has an energy $E_{1,2}^{g=\infty} = \frac{7}{3}\mathcal{E}_0$. For the non-integrable counterpart, the $g = \infty$ spectrum is non-analytic, including the ground state. In both cases, the energy levels are doubly degenerate in $g \to \infty$ limit.

As we mentioned above, the Liu-Qi-Zhang-Chen system is Asymmetric Bethe Ansatz integrable; it inherits additional integrals of motion of the underlying $\tilde{G}_2$ kaleidoscope (see Fig. 5). Thanks to the additional conserved quantities, levels of the same parity with respect to the $180°$ rotation (shown with the same colors) are allowed to cross. This is indeed what we observe in Fig. 6(a). The presence of such crossings is a manifestation of integrability.

In the non-integrable case, most of the crossings are lifted. The crossing $\mathbf{c_5'}$ (see Fig. 6(b)) constitutes a seeming exception. However, a much more accurate calculation reveals an avoided crossing. In the integrable case, this crossing remains a true one.

Explanation for the resilience of some of the crossings involves the interaction-insensitive eigenstates of the Liu-Qi-Zhang-Chen system, marked by $\mathbf{f}$ in Fig. 6(a). These states have a node along the interaction line. When the integrability is broken, the remnants of this node (marked $\mathbf{f'}$) survive, leading to a reduced coupling to other eigenstates.

In particular, the state $\mathbf{f_{III}'}$, non-integrable counterpart of the state $\mathbf{f_{III}}$ of the integrable system is responsible for the existence of the near-crossing $c_5'$ (see Fig. 6(b)). The Fig. 7 provides more details. This state is a linear combination of the three eigenstates of the noninteracting system (the black-walled rectangular $L \times (\sqrt{3}L)$ billiard (Fig. 7)), with the quantum numbers $(n_1, n_2) = \{(2, 8), (3, 7), (5, 1)\}$ (see (1)).

At the same time, the state $\mathbf{f_{III}}$ consists of two smoothly connected eigenstates of two similar right triangular $L \times (\sqrt{3}L)$ hard-wall billiards (the yellow-walled triangular billiard and its copy (Fig. 7)), with the quantum numbers $(\tilde{n}_1, \tilde{n}_2) = (1, 7)$. It is the even—with

respect to a 180° rotation about the alcove center—eigenstate of the Liu-Qi-Zhang-Chen system at $g = \infty$; its energy spectrum is given by (2).

Finally, the state represented in this Figure is a tiling of six smoothly connected eigenstates of of six similar $L/\sqrt{3} \times L$ (the green-walled triangular billiard[2] and its five copies (Fig. 7)), with the quantum numbers $(\tilde{\tilde{n}}_1, \tilde{\tilde{n}}_2) = (2, 3)$. Any eigenstate of the green-walled billiard can be used to generate an interaction-insensitive eigenstate of the Liu-Qi-Zhang-Chen system. Hence, in general, the energies of the interaction-insensitive eigenstates will be given by

$$
\begin{aligned}
E^{g\text{-insensitive}}_{\tilde{\tilde{n}}_1, \tilde{\tilde{n}}_2} &= \left(\tilde{\tilde{n}}_1^2 + \tilde{\tilde{n}}_1 \tilde{\tilde{n}}_2 + \tilde{\tilde{n}}_2^2\right) \mathcal{E}_0 \\
\tilde{\tilde{n}}_2 &> \tilde{\tilde{n}}_1 \geq 1
\end{aligned}
\tag{3}
$$

The node along the interaction line make the state energy be insensitive to the interactions. Deviations from the integrable mass ratio $\beta = 3$ will eventually populate the interaction line. However, the extremely weak splitting at the point $\mathbf{c'_5}$ in Fig. 5(b) indicates that even at $\beta = 3.58\ldots$ some traces of integrability at $\beta = 3$ remain. In particular, as we found numerically, the presence of near-crossings at $\beta \neq 3$ prevents formation of the zero-energy hole in the level spacing distribution making the deviations from integrability difficult to detect.

## 2.3 Asymmetric Bethe Ansatz: general considerations

The ABA method can be extended to any pair formed by a reflection group $\mathcal{G}$ and its reflection subgroup $\mathcal{H}$. Reflection subgroups of the reflection groups have been studied in detail in Ref [10]. Results of this article can be summarized as follows:

- Reflection subgroups of the finite reflection groups are listed in Chapter 3 of Ref. [10].

- For any undecomposable affine reflection group $\mathcal{G}$ and for any integer scaling factor $d$, a reflection subgroup $\mathcal{H}$ can be found whose alcove is a homothetic copy of the alcove of $\mathcal{G}$, with a dilation factor $d$ (Lemma 9 of [10], see Fig. 8(a) as an example).

- If a group $\mathcal{G}$ is decomposable onto a product of smaller reflection groups, the corresponding reflection subgroup $\mathcal{H}$ can be a product of dilation subgroups of the components of $\mathcal{G}$, with, generally, different dilation factors (Fig. 8(b)).

- Affine reflection groups $\mathcal{G} = \tilde{C}_2, \tilde{G}_2, \tilde{F}_4$ have reflection subgroups $\mathcal{H}$ with alcoves that are similar to the ones for $\mathcal{G}$ but not homothetic to them (Table 2 of Ref. [10], see Fig. 8(c) as an example).

- There are several potentially important cases where the alcove of $\mathcal{H}$ is not similar to the one of $\mathcal{G}$. $\mathcal{H}$ can be either an indecomposable (Table 3 of Ref. [10]) or a decomposable reflection subgroup (Table 5 of Ref. [10], see Fig. 8(d) as an example; also this is the example that solves the Liu-Qi-Zhang-Chen problem [8][3]). Here $\mathcal{G}$ is an indecomposable affine reflection group in this context.

- An alcove of $\mathcal{H}$ may have an infinite volume (Theorem 3 of Ref. [10], see Fig. 8(e) as an example).

---

[2]This billiard is also an alcove for the reflection group $\tilde{G}_2$ that is used to generate solutions of the Liu-Qi-Zhang-Chen problem.

[3]The kaledoscopes of Fig. 8(d) and Fig. 5(a) are mirror images of each other, with respect to a mirror at 45° to the horizon.

- All the techniques described in Ref. [10] may be combined with one another. Figure 8(f) shows an example of a combination of a non-similar subgroup with a decomposable homothety.

# 3  Summary and outlook

In this article, we generalize the conventional Bethe Ansatz, by breaking the symmetry $\mathcal{G}$ of the generalized kaleidoscope the Ansatz is based on. The new method allows to replace a set of semitransparent mirrors of the original generalized kaleidoscope by reflecting mirrors. The set of such modified mirrors must coincide with mirrors of a reflection subgroup $\mathcal{H}$ of the original reflection group $\mathcal{G}$. One of the consequences of this generalization is that the mirrors crossing at an angle $\pi/$(odd number) are no longer required to have the same coupling constant. We decided to call the new exact solution method the Asymmetric Bethe Ansatz (ABA).

The next step will be to identify the *previously unknown, empirically relevant instances of the ABA*, besides the Liu-Qi-Zhang-Chen that our paper studied in detail. One already known instance is the problem of scattering of two one-dimensional $\delta$-interacting bosons on a $\delta$-barrier: it was found to be solvable in the spatially odd sector of the Hilbert space [11]. There, $\mathcal{G} = C_1$ and $\mathcal{H} = A_1$ represented by a mirror along the second diagonal (see Ref. [10], Lemma 2, Corollary 1). Another known example is $\mathcal{G} = \tilde{A}_{N-1}$ and $\mathcal{H} = \tilde{A}_{N_1-1} \times \tilde{A}_{N_2-1}$, with $N_1 + N_2 = N$ (where $\tilde{A}_1$ is associated with $\tilde{I}_1$, and $\tilde{A}_0$ is associated with the trivial group). It can be regarded as a $N_1$ spin-up/$N_2$ spin-down sector of the Yang-Gaudin model of spin-1/2 one-dimensional $\delta$-interacting fermions [6,7], on a ring; it's bosonized version has been studied in Ref. [12]. This case is covered by Theorem 3 of Ref. [10]. Figure 8(e) provides an example for $N = 3 = 1 + 2$. Additionally, the ($\mathcal{G} = \tilde{C}_N$, $\mathcal{H} = \tilde{C}_{N_1} \times \tilde{C}_{N_2}$ assignment leads to the Yang-Gaudin model in a hard-wall box (Table 5 of Ref. [10]).

# Acknowledgements

We are immeasurably grateful to Vanja Dunjko, Anna Minguzzi, Patrizia Vignolo, and Yunbo Zhang for numerous discussions.

**Funding information**   M.O. was supported by the NSF Grant No. PHY-1912542. G.E.A. acknowledges support by the Spanish Ministerio de Ciencia e Innovación (MCIN/AEI/ 10.13039/501100011033, grant PID2020-113565GB-C21), and by the Generalitat de Catalunya (grant 2021 SGR 01411).

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

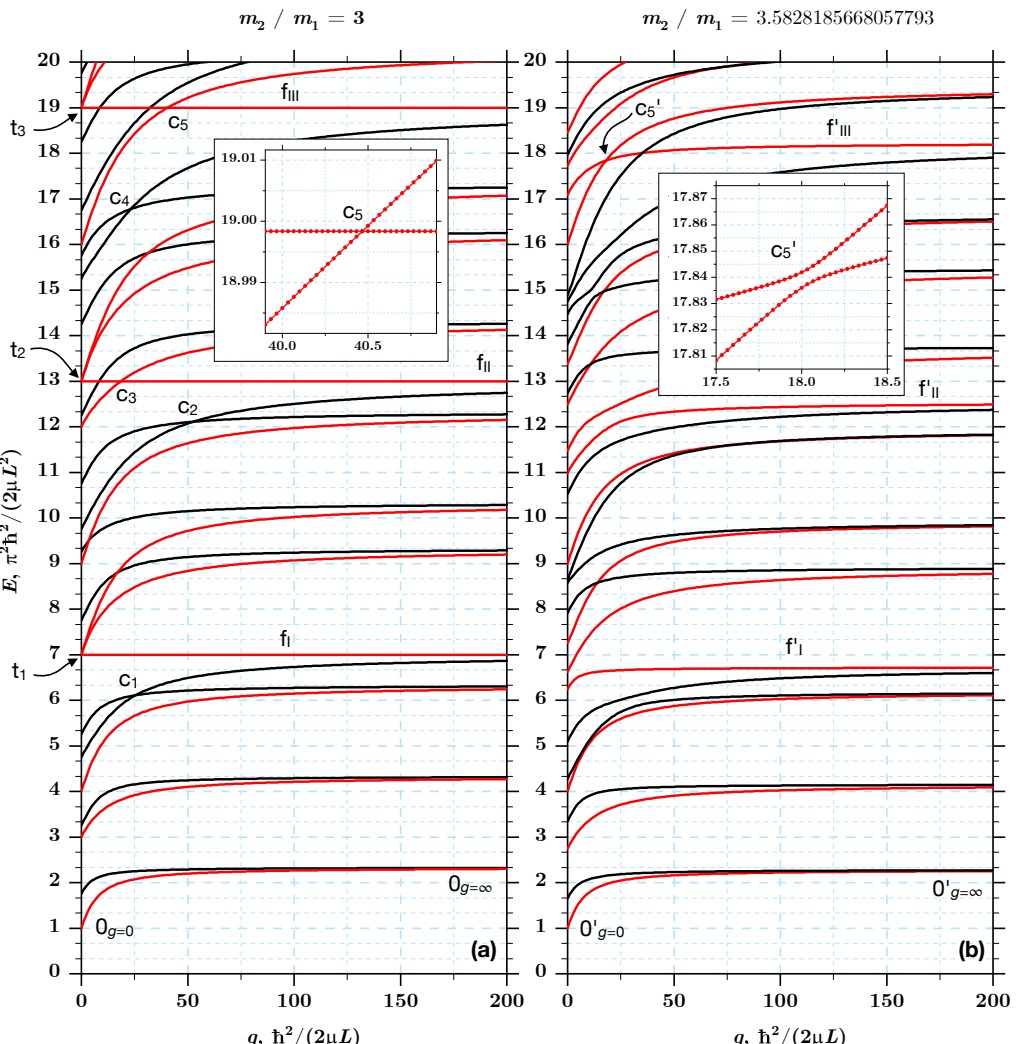

Figure 6: **Energy levels as a function of the coupling constant, for two $\delta$-interacting particles in a hard-wall box of length $L$.** (a) Mass ratio $m_1$:$m_2$ is 1:3 for Subfigure (a) and 1:3.5828185668057793... for Subfigure (b). The former corresponds to the Liu-Qi-Zhang-Chen system represented in Fig. 4. Red(black) lines correspond to the states that are even(odd) under the 180° rotation about the center of a rectangular billiard similar to the one presented in Fig. 4. Labels $\mathbf{0_{g=0}}$ and $\mathbf{0'_{g=0}}$ mark the non-interacting integrable and non-integrable ground state energies respectively. Labels $\mathbf{0_{g=\infty}}$ and $\mathbf{0'_{g=\infty}}$ denote the integrable and non-integrable ground state energies in the hard-core regime respectively. The crossings between the same-parity energy levels are marked with $\mathbf{t}$ and $\mathbf{c}$ labels. The $\mathbf{t}$ crossings are triply-degenerate levels of the noninteracting system, and the interpretation of this degeneracy goes beyond the scope of this paper. The $\mathbf{c}$ crossings appear at finite values of coupling. Insets in (a) and (b) magnify the regions in the vicinity of $\mathbf{c_5}$ and $\mathbf{c'_5}$ points respectively. In spite of its appearance, the crossing $\mathbf{c'_5}$ is indeed an avoided crossing, while the $\mathbf{c_5}$ is a true one, as expected from the integrability considerations. The energy levels $\mathbf{f}$ and $\mathbf{f'}$ are the interaction-insensitive and near-interaction-insensitive eigenstates of the integrable and non-integrable systems respectively.

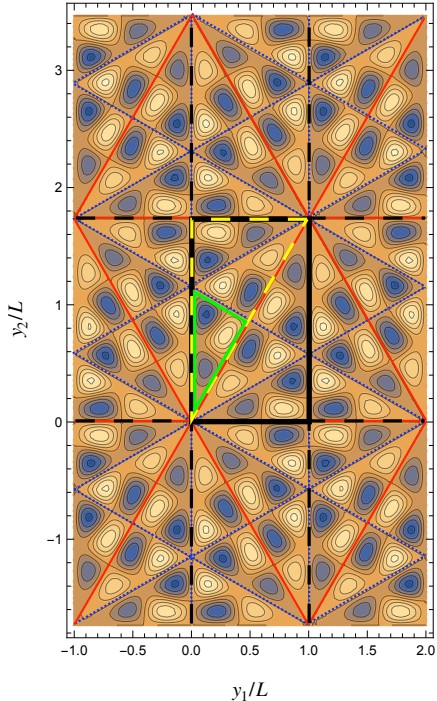

Figure 7: **An example of an interaction-insensitive eigenstate of the Liu-Qi-Zhang-Chen system.** In Fig. 6(a)), the energy of this state is marked by $\mathbf{f'_{III}}$. This state explains the resilience of the level crossing $\mathbf{c'_5}$ in Fig. 6(b). Any interaction-insensitive state is a smooth sign-alternating tiling of the six identical eigenstates of the green-walled billiard. Any eigenstate of the green-walled billiard can be used to generate this tiling. (Curiously, these eigenstates are also the eigenstates of the hard-core version of the conventional generalized kaleidoscope $\tilde{G}_2$ that was used to generate solutions of the Liu-Qi-Zhang-Chen problem (see Fig. 5(a)).) Because of the absence of cusp, this tiling is naturally an eigenstate of the free system. Because of the node along the interaction line (red), adding interaction does not affect the eigenstate. As expected, a tiling of three out of six green-walled billiard eigenstates is also an eigenstates of the yellow-walled billiard relevant to the hard-core interaction case.

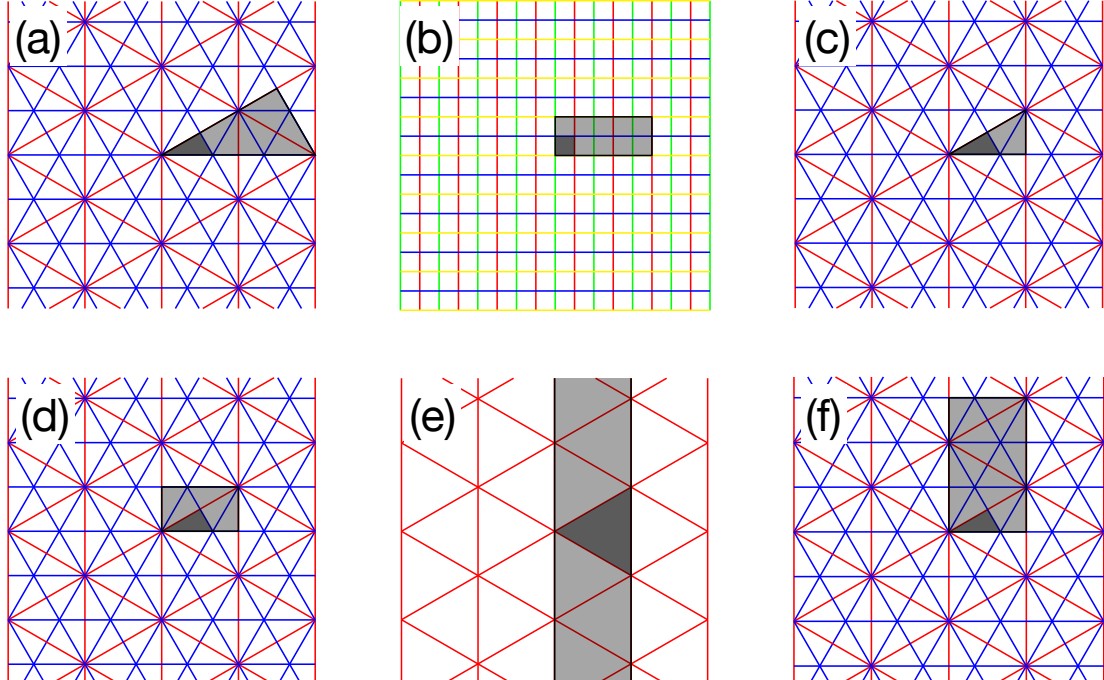

Figure 8: **Several significant features of the complete list of reflection subgroups $\mathcal{H}$ of indecomposable reflection groups $\mathcal{G}$ presented in [10].** Here, a deep-grey polygon represents a sample alcove of the corresponding indecomposable reflection groups $\mathcal{G}$; a light-grey polygon is an alcove of the corresponding reflection subgroup $\mathcal{H}$ of $\mathcal{G}$. (a) All indecomposable affine reflection groups have reflection subgroups whose alcoves are homothetic copies of the original group. (b) Decomposable affine reflection groups may several distinct dilution factors. (c) There are several cases when the $\mathcal{H}$ alcove is similar to the $\mathcal{G}$ alcove but not homothetic to it. (d) In several cases, the $\mathcal{H}$ alcove has a shape that is completely different from the shape of the $\mathcal{G}$ alcove. (e) $\mathcal{H}$ alcove may have an infinite volume. (f) Methods for generating the reflection subgroups $\mathcal{H}$ described in [10] can be combined.