# Peer review of "Asymmetric Bethe Ansatz"

_SciPost Physics_

## Round 1 · Referee Report · Anonymous (Referee 1) · 2024-3-12

Strengths

  1. Extension of the method of the Bethe Ansatz.
  2. Connection to results of historical importance.

Weaknesses

  1. Abstract, introduction, and discussion not accessible enough.
  2. Very few concrete examples, very few concrete formulas.
  3. Discussion of other examples not satisfying.

Report

The paper discusses an extension of the Bethe Ansatz method. It goes back to historical work of Michel Gaudin, and now it addresses a new solution of a recently discovered problem. Based on results of Gaudin it seemed that this recent problem can not be integrable, whereas it appears to be. This apparent contradiction is now resolved.

I did not check all computations, but the results are likely correct.
However, I find serious problems with the clarity and accessibility of the text. These problems are given briefly above (weaknesses), and I also formulate some requests below.

I can not judge whether the article can be published in SciPost Physics, or SciPost Core. Only after a major revision can I make a good decision. In the present form, without major revision, this is definitely SciPost Core. However, after a good revision, it could be considered for SciPost Physics. But this will need a reconsideration after major revision.

Requested changes

  1. The abstract and the introduction starts with the kaleidoscopes and the mirrors. For a very selected set of readers this is clear, but even for the average integrability expert this is not immediately obvious. I find it problematic, that the actual two body or many body perspective is given only in Sections 1.2 and 1.3 of the Introduction, and Section 1.1 is very abstract. This should be re-ordered somehow. If the authors desire publication in SciPost Physics, then first there should be the general context of many-body integrability (delta function interacting gases), and only afterwards the abstract framework with the mirrors. The same holds true for the abstract: there should be a way to present those sentences in a less abstract way. After all, the Liu-Zhang-Chen problem is a concrete and immediately understandable problem.

  2. I see it as a big problem that there is not a single Bethe wave function in the text. It seems that the idea of the authors is to build on all the previous literature, but I find this as a bad choice. My point of view is that in the specific case of the Liu-Zhang-Chen problem the final Bethe wave function should be given explicitly. And maybe also for some other examples. Also, a more direct explanation should be given for the connection of the Liu-Zhang-Chen problem and the present treatment. If I understand it correctly, it is only a rescaling of one of the coordinates. Nevertheless I believe that this point should be made more concrete and explicit.

  3. For me it was not clear: How many particles can we have for the Asymmetric B.A. I guess this is somehow explained in the text, or it should be clear from the derivations. But from a first read it was not clear to me. The Liu-Zhang-Chen problem has two particles in 1D, or equivalently, one particle in 2D. To what extent can we increase the number of particles or the dimensionality? There is one sentence about Figure 8e, which should be 3 particles, if I understand correctly? This is way too quick, and more explanation should be given. It should be very clear: which are the examples which correspond to some physically meaningful delta-function interacting model, and which are some other exotic models, which do not correspond to experimentally relevant situations?

  4. The abbreviation ABA is not the best. For most people ABA means Algebraic Bethe Ansatz. I suggest to use something else. AsymBA or AsBA, or something else.

---

## Round 1 · Referee Report · Anonymous (Referee 2) · 2024-4-19

Strengths

-Asymmetric Bethe Ansatz extends the known class of Bethe-Ansatz solvable models
-Asymmetric Bethe Ansatz explains the integrability of the Liu-Qi-Zhang-Chen system

Weaknesses

-Absence of any introduction/context for non-specialists. The manuscript is fully accessible only by a very narrow community working on similar topics

-A connection of these results with exactly solvable models in statistical physics and/or in quantum gases is lacking or, at least, it is unsatisfactory.

Report

The manuscript discusses a very specific problem: the integrability of the Liu-Qi-Zhang-Chen system cannot be explained with the conventional Bethe-Ansatz approach. To resolve this apparent contradiction, the authors re-interpret the integrability of the system in terms of generalized kaleidoscopes (cf. M. Gaudin) and manage to relax some of the standard integrability conditions. This leads to the concept of “Asymmetric Bethe Ansatz” solvability, of which the Liu-Qi-Zhang-Chen system is a particular instance.

I can understand the stylistic choice made by the authors: the statement of the problem and the solution to it in the manuscript is made clear from the first sentences of the abstract. However, I find it extremely disappointing that any other kind of (even qualitative) understanding of the authors’ results is very hard to get for a non-specialized audience.

In my opinion, a full understanding of the manuscript is possible only by a very narrow audience that is already knowledged in the specific field. After quite some effort to understand, I believe the authors’ results are correct, although I was unable to follow all the details of their derivation.

I believe that this is the major issue of this manuscript. The potential interest of these results (cf strengths) for a broad community working on integrable models (e.g. in connection to statistical models and/or experiments with ultracold atoms and ions) has to overcome this technical barrier, making de-facto this study of very limited impact in its actual format.

To improve the readability of the paper, I strongly encourage the authors to:

-Re-think the abstract, possibly avoiding technicalities, engaging a broader community in presenting a new method for finding new solvable models.

-In Introduction — a general introduction is currently lacking. The specific problem should be placed into a more general context, e.g. from formal and mathematical aspects of integrability to statistical physics problems and cold atoms experiments.

-In Sec.1.2 — Add examples relating to the (standard) Gaudin formulation of integrability in terms of generalized kaleidoscopes to (coordinate) Bethe ansatz. For instance, explaining the integrability of the delta-interacting Bose gas could introduce non-specialists to this new formalism.

-In Sec.2.2— I would give more space to commenting on some concrete physical setting (e.g. Ref[11], [12]) where the Asymmetric Bethe ansatz could explain an “empiric manifestation” of integrability. This should be explained in simple terms, possibly using Hamiltonians, wavefunctions, and a language that is accessible to a broad audience.

In conclusion, I cannot recommend the publication of the manuscript in SciPost journals in its current form. I suggest to reconsider the manuscript for publication in SciPost Physics after a revision by the authors according to the points abovementioned.

Requested changes

(see report)

Recommendation

Ask for major revision

---

## Editorial Decision

resubmitted